# Using a one dimensional parabolic model of the full-batch loss to estimate learning rates during training

## Abstract

A fundamental challenge in Deep Learning is to find optimal step sizes for stochastic gradient descent automatically. In traditional optimization, line searches are a commonly used method to determine step sizes. One problem in Deep Learning is that finding appropriate step sizes on the full-batch loss is unfeasibly expensive. Therefore, classical line search approaches, designed for losses without inherent noise, are usually not applicable. Recent empirical findings suggest that the full-batch loss behaves locally parabolically in the direction of noisy update step directions. Furthermore, the trend of the optimal update step size changes slowly. By exploiting these findings, this work introduces a line-search method that approximates the full-batch loss with a parabola estimated over several mini-batches. Learning rates are derived from such parabolas during training. In the experiments conducted, our approach mostly outperforms SGD tuned with a piece-wise constant learning rate schedule and other line search approaches for Deep Learning across models, datasets, and batch sizes on validation and test accuracy.

## 1 Introduction

Automatic determination of an appropriate and loss function-dependent learning rate schedule to train models with stochastic gradient descent (SGD) or similar optimizers is still not solved satisfactorily for Deep Learning tasks. The long-term goal is to design optimizers that work out-of-the-box for a wide range of Deep Learning problems without requiring hyper-parameter tuning. Therefore, although well-working hand-designed schedules such as *piece-wise constant learning rate* or *cosine decay* exist (see (Loshchilov & Hutter, 2017; Smith, 2017)), it is desired to infer problem depended and better learning rate schedules automatically.

This work builds on recent empirical findings; among those are that the full-batch loss tends to have a simple parabolic shape in SGD update step direction (Mutschler & Zell, 2021; 2020) (see Figure 1) and that the trend of the optimal update step changes slowly (Mutschler & Zell, 2021) (see Figure Figure 2). Exploiting these and more found observations, we introduce a line search approach, approximating the full-batch loss along lines in SGD update step direction with parabolas. One parabola is sampled over several batches to obtain a more exact approximation of the full-batch loss. The learning rate is then derived from this parabola. As the trend of the optimal update step-size on the full-batch loss changes slowly, the line search only needs to be performed occasionally; usually, every 1000th step. This results in low computational overhead.

The major contribution of this work is the combination of recent empirical findings to derive a line search method, which is built upon real-world observations and less on theoretical assumptions. This method outperforms the most prominent line search approaches introduced for Deep Learning (Vaswani et al. (2019); Mutschler & Zell (2020); Kafka & Wilke (2019); Mahsereci & Hennig (2017)) across models, datasets usually considered in optimization for Deep Learning, in almost all experiments. In addition, it almost always outperforms SGD tuned with a piece-wise constant learning rate schedule on validation and test accuracy. The second important contribution is that we are the first to analyze how the considered line searches perform under high gradient noise that originates from small batch sizes. While all considered line searches perform poorly -mostly because they rely on mini-batch losses-, our approach adapts well to increasing gradient noise by approximating the full-batch loss. However, a significant performance gap still exists between our

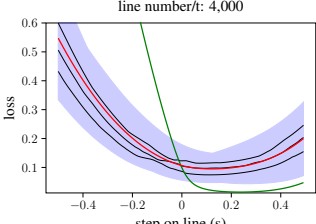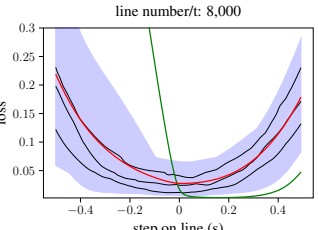

Figure 1: Losses along the lines of the SGD training processes exhibit a parabolic shape. The loss of the direction defining mini-batch (green) is excluded from the distribution of mini-batch losses to show that it is significantly different. This makes line searches on it unfavorable. In addition, the parabolic property articulates stronger for the full-batch loss (red); thus, this work aims to approximate it efficiently with a parabola. The figure is created with code and data from Mutschler & Zell (2021).

line search and an optimally tuned SGD optimizer using momentum, weight decay, and a problem-specific hand-designed learning rate schedule.

The paper is organized as follows: Section 2 provides an overview of related work. Section 3 derives our line search approach and introduces its mathematical and empirical foundations in detail. In Section 4 we analyze the performance of our approach across datasets, models, and gradient noise levels. Also, a comprehensive hyper-parameter, runtime, and memory consumption analysis is performed. Finally, we end with discussion including limitations in Sections 5 & 6.

## 2 RELATED WORK

**Deterministic line searches:** According to (Jorge & Stephen, 2006, §3), line searches are considered a solved problem, in the noise-free case. However, such methods are not robust to gradient and loss noise and often fail in this scenario since they shrink the search space inadequately or use too inexact approximations of the loss. (Jorge & Stephen, 2006, §3.5) introduces a deterministic line search using parabolic and cubic approximations of the loss, which motivated our approach.
**Line searches on mini-batch and full-batch losses and why to favor the latter.** The following motivates the goal of our work to introduce a simple, reasonably fast, and well-performing line-search approach that approximates full-batch loss.
Many exact and inexact line search approaches for Deep Learning are applied on mini-batch losses (Mutschler & Zell, 2020; Berrada et al., 2020; Rolinek & Martius, 2018; Baydin et al., 2018; Vaswani et al., 2019). (Mutschler & Zell, 2020) approximates an exact line search by estimating the minimum of the mini-batch loss along lines with a one-dimensional parabolic approximation. The other approaches perform inexact line searches by estimating positions of the mini-batch loss along lines, which fulfill specific conditions. Such, inter alia, are the Goldberg, Armijo, and Wolfe conditions (see Jorge & Stephen (2006)). For these, convergence on convex stochastic functions can be assured under the interpolation condition (Vaswani et al., 2019), which holds if the gradient with respect to each batch converges to zero at the optimum of the convex function. Under this condition, the convergence rates match those of gradient descent on the full-batch loss for convex functions (Vaswani et al., 2019). However, relying on those assumptions and on mini-batch losses only does not lead to robust optimization, especially not if the gradient noise is high, as will be shown in Section 4. (Mutschler & Zell, 2021; 2020) even showed that exact line searches on mini-batch losses are not working at all. Line searches on the non-stochastic full-batch loss show linear convergence on any deterministic function that is twice continuously differentiable, has a relative minimum, and only positive eigenvalues of the Hessian at the minimum (see Luenberger et al. (1984)). In addition, they are independent of gradient noise. Therefore, it is reasonable to consider line searches on the full-batch loss. However, these are cost-intensive since a massive amount of mini-batch losses for multiple positions along a line must be determined to measure the full-batch loss.
Probabilistic Line Search (PLS) (Mahsereci & Hennig, 2017) addresses this problem by performing Gaussian Process Regressions, which result in multiple one-dimensional cubic splines. In addition, a probabilistic belief over the first (aka Armijo condition) and second Wolfe condition is introduced to find appropriate update positions. The major drawback of this conceptually appealing method is

Figure 2: Several metrics to compare update step strategies on the full-batch losses along 10,000 lines measured by Mutschler & Zell (2021): 1. update step sizes, 2. accumulated loss improvement per step given as: $l(0) - l(s_{upd})$ where $s_{upd}$ is the update step of a specific optimizer. This is the locally optimal improvement to the minimum of the full-batch loss along a line. The right plot shows almost proportional behavior between the optimal update step and the negative gradient norm of the direction defining mini-batch loss. The LABPAL&SGD version of our approach performs almost optimal on ground truth data. Results LABPAL&NSGD are almost identical and thus omitted. Plotting code based on Mutschler & Zell (2021) with addition of our proposed approach.

its high complexity and slow training speed. A different approach working on the full-batch loss is Gradient-only line search (GOLSI) (Kafka & Wilke, 2019). It approximates a line search on the full-batch loss by considering consecutive noisy directional derivatives whose noise is considerably smaller than the noise of the mini-batch losses. In practice, its performance is rather weak.

**Empirical properties of the loss landscape:** In Deep Learning, loss landscapes of the true loss (over the whole distribution), the full-batch loss, and the mini-batch loss can, in general, be highly non-convex. However, to efficiently perform a line search, some properties of these losses have to be apparent. Little is known about such properties from a theoretical perspective; however, several works suggest that loss landscapes tend to be simple and have some properties: Mutschler & Zell (2021); Li et al. (2018); Xing et al. (2018); Mutschler & Zell (2020); Chae & Wilke (2019); Mahsereci & Hennig (2017); Goodfellow et al. (2015); Fort & Jastrzebski (2019); Draxler et al. (2018). (Mahsereci & Hennig, 2017; Xing et al., 2018; Mutschler & Zell, 2021; 2020) suggest that the full-batch loss $\mathcal{L}$ along lines in negative gradient directions tend to exhibit a simple shape for a set of Deep Learning problems. This set includes at least classification tasks on CIFAR-10, CIFAR-100, and ImageNet. (Mutschler & Zell, 2021) sampled the full-batch loss along the lines in SGD update step directions. This was done for $10,000$ consecutive SGD update steps of a ResNet18's training process on a subset of CIFAR-10. Representative plots of their $10,000$ measured full-batch losses along lines are presented in Figure 1. Relevant insights and found properties of these works will be introduced and exploited to derive our algorithm in Section 3.

**Using the batch size to tackle gradient noise:** Besides decreasing the learning rate, increasing the batch size remains an important choice to tackle gradient noise. McCandlish et al. exploits empirical information to predict the largest piratical batch size over datasets and models. De et al. adaptively increases the batch size over update steps to assure that the negative gradient is a descent direction. Smith & Le introduces the *noise scale*, which controls the magnitude of the random fluctuations of consecutive gradients interpreted as a differential equation. The latter leads to the observation that increasing the batch size has a similar effect as decreasing the learning rate (Smith et al., 2018), which is exploited by our algorithm.

## 3 OUR APPROACH: LARGE-BATCH PARABOLIC APPROXIMATION LINE SEARCH (LABPAL)

### 3.1 MATHEMATICAL FOUNDATIONS

In this subsection, we introduce the mathematical background relevant for line searches and challenges that must be solved in order to perform line searches in Deep Learning.

We consider the problem of minimizing the full-batch loss $\mathcal{L}$, which is the mean over a large amount of sample losses $L$:

$$\mathcal{L} \; : \; \mathbb{R}^n \to \mathbb{R}, \; \theta \mapsto \frac{1}{|\mathbb{D}|} \sum_{d \in \mathbb{D}} L_d(\theta), \tag{1}$$

where $\mathbb{D}$ is a finite dataset and $\theta$ are $n$ parameters to optimize. To increase training speed generally, a mini-batch loss $\mathcal{L}_\mathbb{B}$, which is a noisy estimate of $\mathcal{L}$, is considered:

$$\mathcal{L}_\mathbb{B} \; : \; \mathbb{R}^n \to \mathbb{R}, \; \theta \mapsto \frac{1}{|\mathbb{B}|} \sum_{d \in \mathbb{B} \subset \mathbb{D}} L_d(\theta), \tag{2}$$

with $|\mathbb{B}| \ll |\mathbb{D}|$. We define the mini-batch gradient at step $t$ as $\mathbf{g}_{\mathbb{B},t} \in \mathbb{R}^n$ as $\nabla_{\theta_t} \mathcal{L}_\mathbb{B}(\theta_t)$.

For our approach, we need the full-batch loss along the direction of the negative normalized gradient of a specific mini-batch loss. At optimization step $t$ with current parameters $\theta_t$ and a direction defining batch $\mathbb{B}_t$, $\mathcal{L}_\mathbb{B}$ along a line with origin $\theta_t$ in the negative direction of the normalized batch gradient $\hat{\mathbf{g}}_{\mathbb{B},t} = \mathbf{g}_{\mathbb{B},t}/||\mathbf{g}_{\mathbb{B},t}||$ is given as:

$$l_{\mathbb{B},t} \; : \; \mathbb{R} \to \mathbb{R}, \; s \mapsto \mathcal{L}_\mathbb{B}(\theta_t + s \cdot -\hat{\mathbf{g}}_{\mathbb{B}_t,t}), \tag{3}$$

where $s$ is the step size along the line. The corresponding full-batch loss along the same line is given by:

$$l_t : \; \mathbb{R} \to \mathbb{R}, \; s \mapsto \mathcal{L}(\theta_t + s \cdot -\hat{\mathbf{g}}_{\mathbb{B}_t,t}). \tag{4}$$

Let the step size to the first encountered minimum of $l_t$ be $s_{min,t}$.

Two major challenges have to be solved in order to perform line searches on $\mathcal{L}$:

1. To measure $l_t$ exactly it is required to determine every $L_d(\theta_t + s \cdot -\hat{\mathbf{g}}_{\mathbb{B}_t,t})$ for all $d \in \mathbb{D}$ and for all step sizes $s$ on a line.

2. To assure convergence line searches have to be performed in a descent direction (De et al., 2016). The simplest form is the direction of steepest descent (Luenberger et al., 1984). Therefore, the full-batch gradient $\nabla_\theta \mathcal{L} \; : \; \mathbb{R}^n \to \mathbb{R}^n, \; \theta \mapsto \frac{1}{|\mathbb{D}|} \sum_{d \in \mathbb{D}} \nabla L_d(\theta_t)$ has to be approximated.

To be efficient, $l_t$ has to be approximated sufficiently well with as little data points $d$ and steps $s$ as possible, and one has to use as little $d$ as possible to approximate $\nabla_\theta \mathcal{L}$ approximated sufficiently well. Such approximations are highly dependent on properties of $\mathcal{L}$. Due to the complex structure of Deep Neural Networks, little is known about such properties from a theoretical perspective. Thus, we fall back to empirical properties.

### 3.2 Deriving the algorithm

In the following, we derive our line search approach on the full-batch loss by iteratively exploiting empirically found observations of (Mutschler & Zell, 2021) and solving the challenges for a line search on the full-batch loss (see Section 3.1). Given default values are inferred from a detailed hyper parameter analysis (Section 4.4)

**Observation 1:** *Minima of $l_{\mathbb{B},t}$ can be at significantly different points than minima of $l_t$ and can even lead to update steps, which increase $\mathcal{L}$* (Figure 2 center, green and red curve).

**Derivation Step 1:** This consolidates that line searches on a too low mini-batch loss are unpromising. Consequently, we concentrate on a better way to approximate $l_t$.

**Observation 2:** *$l_t$ can be approximated with parabolas of positive curvature, whose fitting errors are of less than $0.6 \cdot 10^{-2}$ mean absolute distance* (exemplarily shown in Figure 1).

**Derivation Step 2:** We approximate $l_t$ with a parabola ($l(s)_t \approx a_t s^2 + b_t s + c_t$ with $a_t > 0$). A parabolic approximation needs three measurements of $l_t$. However, already computing $l_t$ for one $s$ only is computationally unfeasible. Assuming i.i.d sample losses, the standard error of $l_{\mathbb{B},t}(s)$, decreases with $1/\sqrt{|\mathbb{B}|}$. Thus, $l_{\mathbb{B},t}$ -with a reasonable large batch size- is already a good estimator for the full-batch loss parabola. Consequently, we approximate $l_t$ with $l_{\mathbb{B}_a,t}$ by averaging over multiple $l_{\mathbb{B}_i,t}$ measured with multiple inferences. Thus, the approximation batch size $\mathbb{B}_a$, is significantly larger as the, by GPU memory limited, possible batch size $\mathbb{B}_i$. In our experiments, $\mathbb{B}_a$ is usually chosen to be 1280, which is 10 times larger as $\mathbb{B}_i$. In detail, we measure $l_{\mathbb{B}_a,t}$ at the points $s = 0, 0.0001$ and $0.01$, then we simply infer the parabola's parameters and the update step to the minimum. These values of $s$ empirically lead to the best and numerically most stable approximations.

**Observation 3:** *The trend of $s_{min,t}$ of consecutive $l_t$ changes slowly and consecutive $l_t$ do not change locally significantly.* (Figure 2 left, red curve).

**Observation 4:** *$s_{min,t}$ and the direction defining batch's $||\mathbf{g}_{\mathbb{B}_t,t}||$ are almost proportional during training.* (Figure 2 right).

**Derivation Step 3:** Using measurements of $l_{\mathbb{B}_a,t}$ to approximate $l_t$ with a parabola is by far to slow to compete against SGD if done for each weight update. By exploiting Observation 3 we can approximate $l_t$ after a constant amount of steps and reuse the measured learning rate $\lambda$ or update step size $s_{upd}$ for subsequent steps. In this case, $\lambda$ is a factor multiplied by $\mathbf{g}_{\mathbb{B},t}$, whereas $s_{upd}$ is a factor multiplied by $\hat{\mathbf{g}}_{\mathbb{B},t}$. Observation 4 allows us to reuse $\lambda$. In our experiments, it is sufficient to measure a new $\lambda$ or $s_{upd}$ every 1000 steps only.

**Derivation Step 4:** So far, we can approximate $l_t$ efficiently and, thus, overcome the first challenge (see Section 3.1). Now, we will overcome the second challenge; approximating the full batch loss gradient for each weight update step:

For this, we revisit Smith et al. (2018) who approximates the magnitude of random gradient fluctuations, that appear if training with a mini-batch gradient, by the *noise scale* $\nu \in \mathbb{R}$:

$$\nu \approx (\lambda|\mathbb{D}|)/|\mathbb{B}|, \tag{5}$$

where $\lambda$ is the learning rate, $|\mathbb{D}|$ the dataset size and $|\mathbb{B}|$ the batch size. If the random gradient fluctuations are reduced, the approximation of the gradient gets better. Since we want to estimate the learning rate automatically, the only tunable parameter to reduce the *noise scale* is the batch size.

**Observation 5:** *The variance of consecutive $s_{min,t}$ is low, however, it increases continuously during training* (Figure 2 left, red curve).

**Derivation Step 5:** It stands to reason that the latter happens because the random gradient fluctuations increase. Consequently, during training, we increase the batch size for weight updates by iteratively sampling a larger batch with multiple inferences. This reduces the variance of consecutive $s_{min,t}$ and lets us reuse estimated the $\lambda$ or $s_{upd}$ for more steps. After experiencing unusable results with the approach of (De et al., 2016) to determine appropriate batch sizes, we stick to a simple piece-wise constant batch size schedule doubling the batch size after two and after three-quarters of the training.

**Observation 6:** *On a global perspective a $s_{upd}$ that overestimates $s_{min,t}$ optimizes and generalizes better.*

**Derivation Step 6:** Thus, after estimating $\lambda$ (or $s_{upd}$) we multiply it with a factor $\alpha \in ]1,2[$:

$$\lambda = \alpha s_{min,t}/||\mathbf{g}_{\mathbb{B},t}||. \tag{6}$$

Note that under out parabolic property, the first wolfe condition $w_1$, which is commonly used for line searches, simply relates to $\alpha$: $w_1 = -\frac{\alpha}{2} + 1$ (see Appendix F).

---

**Algorithm 1** LABPAL&SGD. Simplified conceptional pseudo-code of our proposed algorithm, which estimates update steps on a parabolic approximation of the full-batch loss. See the published source code for technical details. Default values are given in parenthesis. For LABPAL&NSGD SGD is replaced with NSGD, and the update step is measured instead of the learning rate.

**Input:** Hyperparameters:
- initial parameters $\theta_0$
- approximation batch size $|\mathbb{B}_a|$ (1280)
- inference batch size $|\mathbb{B}_i|$ (128)
- SGD steps $n_{\text{SGD}}$ (1000),  # or NSGD steps
- step size adaptation $\alpha > 1$ (1.8)
- training steps $t_{max}$ (150000)
- batch size schedule $k(t) = \begin{cases} 1, & \text{if } t \leq \lfloor t_{max} \cdot 0.5 \rfloor \\ 2, & \text{elif } t \leq \lfloor t_{max} \cdot 0.75 \rfloor \\ 4, & \text{elif } t > \lfloor t_{max} \cdot 0.75 \rfloor \end{cases}$

```
1:  # Variables have global scope
2:  sampledBatchSize ← 0
3:  performedSGDsteps ← 0
4:  learningRate ← 0
5:  θ ← θ_0
6:  state ← 'line search'
7:  direction ← current batch gradient
8:  t ← 0
9:  while t < t_max do
10:     if state is 'line search' then
11:         PERFORM_LINE_SEARCH_STEP()
12:     end if
13:     if state is 'SGDTraining' then
14:         PERFROM_LARGE_BATCH_SGD_STEP()
15:     end if
16: end while
17: return θ
```

```
18: procedure PERFORM_LINE_SEARCH_STEP()
19:     if sampledBatchSize < |B_a| then
20:         update estimate L̂ of L with
            over multiple inferences sampled L_{B_t,t} with
            |B_t| = k(t) · |B_i|)
21:         increase sampledBatchSize by |B_t| and t by k(t)
22:     else
23:         learningRate ← perform parabolic approximation with
            3 values of L̂ along the search direction and estimate the
            learning rate.
24:         learningRate ← learningRate ·α
25:         set sampledBatchSize and performedSGDsteps to 0
26:         state ← 'SGDTraining'
27:     end if
28: end procedure
29:
30: procedure PERFROM_LARGE_BATCH_SGD_STEP()
31:     if performedSGDsteps < n_SGD then
32:         θ ← perform SGD update with learningRate and over
            multiple inferences sampled L_{B_t,t}
            with |B_t| = k(t) · |B_i|)
33:         increase t by k(t)
34:         increase performedSGDsteps by 1
35:     else
36:         direction ← current batch gradient
37:         state ← 'line search'
38:     end if
39: end procedure
```

Combining all derivations leads to our line search named *large-batch parabolic approximation line search* (LABPAL), which is given in Algorithm 1. It samples the desired batch size over multiple inferences to perform a close approximation of the full-batch loss and then reuses the estimated learning rate to train with SGD (LABPAL&SGD), or it reuses the update step to train with SGD with a normalized gradient (LABPAL&NSGD). While LABPAL&SGD elaborates Observation 4, LABPAL&NSGD completely ignores information from $||\mathbf{g}||$.

# 4 EMPIRICAL ANALYSIS

Our two approaches are compared against other line search methods across several datasets and networks in the following. **To reasonably compare different line search methods, we define a *step* as the sampling of a new input batch.** Consequently, the steps/batches that LABPAL takes to estimate a new learning rate/step size are considered, and optimization processes are compared on their data efficiency.

Since most line search approaches are introduced without a momentum term, no momentum terms are used. Note that the base ideas of the introduced line search approaches can be applied upon any direction giving technique such as Momentum, Adagrad (Duchi et al., 2011) or Adam (Kingma & Ba, 2015). Results are averaged over 3 runs.

## 4.1 PERFORMANCE ANALYSIS ON GROUND TRUTH FULL-BATCH LOSS AND PROOF OF CONCEPT

To analyze how well our approach approximates the full-batch loss along lines, we extended the experiments of Mutschler & Zell (2021) by LABPAL. Mutschler & Zell (2021) measured the full-batch loss along lines in SGD update step directions of a training process; thus, this data provides ground truth to test how well the approach approximates the full-batch loss. In this scenario, LAB-PAL&SGD uses the full-batch size to estimate the learning rate and reuses it for 100 steps. No update step adaptation is applied. Figure 2 shows that LABPAL&SGD fits the update step sizes to the minimum of the full-batch loss and performs near-optimal local improvement. The same holds for LABPAL&NSGD.

We now test how our approaches perform in a scenario for which we can assure that the used empirical observations hold. Therefore, we consider the optimization problem of (Mutschler & Zell, 2021) from which all empirical observations were inferred, which is training a ResNet20 on 8% of CIFAR10. $\mathbb{B}_a$ of 1280 is used for both approaches. Learning rates are reused for 100 steps, and $\alpha = 1.8$ is considered. The batch size is doubled after 5000 and 7500 steps. For SGD $\lambda$ is halved after the same steps. A grid search for the best $\lambda$ is performed. Figure 3 shows that LABPAL&NSGD with update step adaptation outperforms SGD, even though 9% of the training steps are used to estimate new update step sizes. This shows that using the estimated learning rates and step sizes leads to better performance than keeping them constant or decaying them with a piece-wise schedule. In-

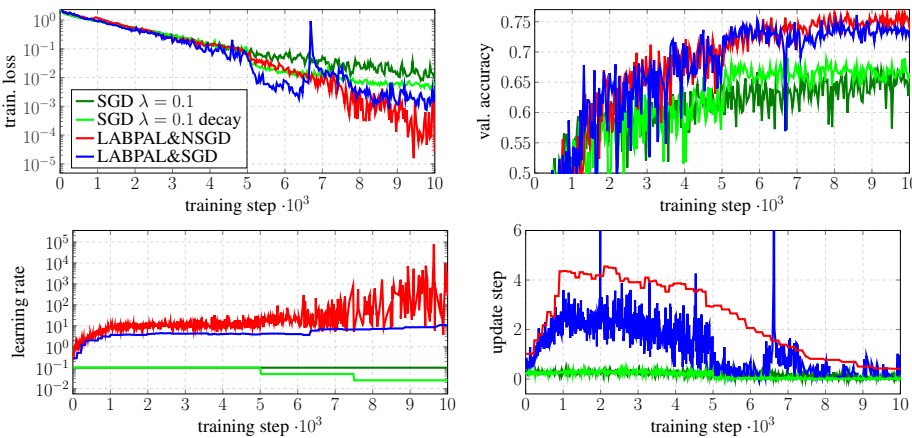

Figure 3: Training process on the problem of which the empirical observations were inferred ( ResNet-20 trained on 8% of CIFAR-10 with SGD). LABPAL&NSGD and LABPAL&SGD outperform SGD. Interestingly LABPAL&NSGD estimated huge $\lambda$s, whereas $s_{upd}$s are decreasing

terestingly huge $\lambda$s of up to $80,000$ are estimated, whereas $s_{upd}$s are decreasing. LABPAL&SGD shows similar performance as SGD; however, it seems beneficial to ignore gradient size information as the better performance of LABPAL&NSGD shows.

## 4.2 Performance comparison to SGD and to other line search approaches

We compare the SGD and NSGD variants of our approach against PLS (Mahsereci & Hennig, 2017), GOLSI (Kafka & Wilke, 2019), PAL (Mutschler & Zell, 2020), SLS (Vaswani et al., 2019) and SGD (Robbins & Monro, 1951). The latter is a commonly used optimizer for Deep Learning problems and can be reinterpreted as a parabolic approximation line search on mini-batch losses (Mutschler & Zell, 2021). PLS is of interest since it approximates the full-batch loss to perform line searches. PAL, GOLSI, SLS on the other hand are line searches optimizing on mini-batch losses directly. For SGD, a piece-wise constant learning rate schedule divides the learning rate after two and again after three-quarters of the training.

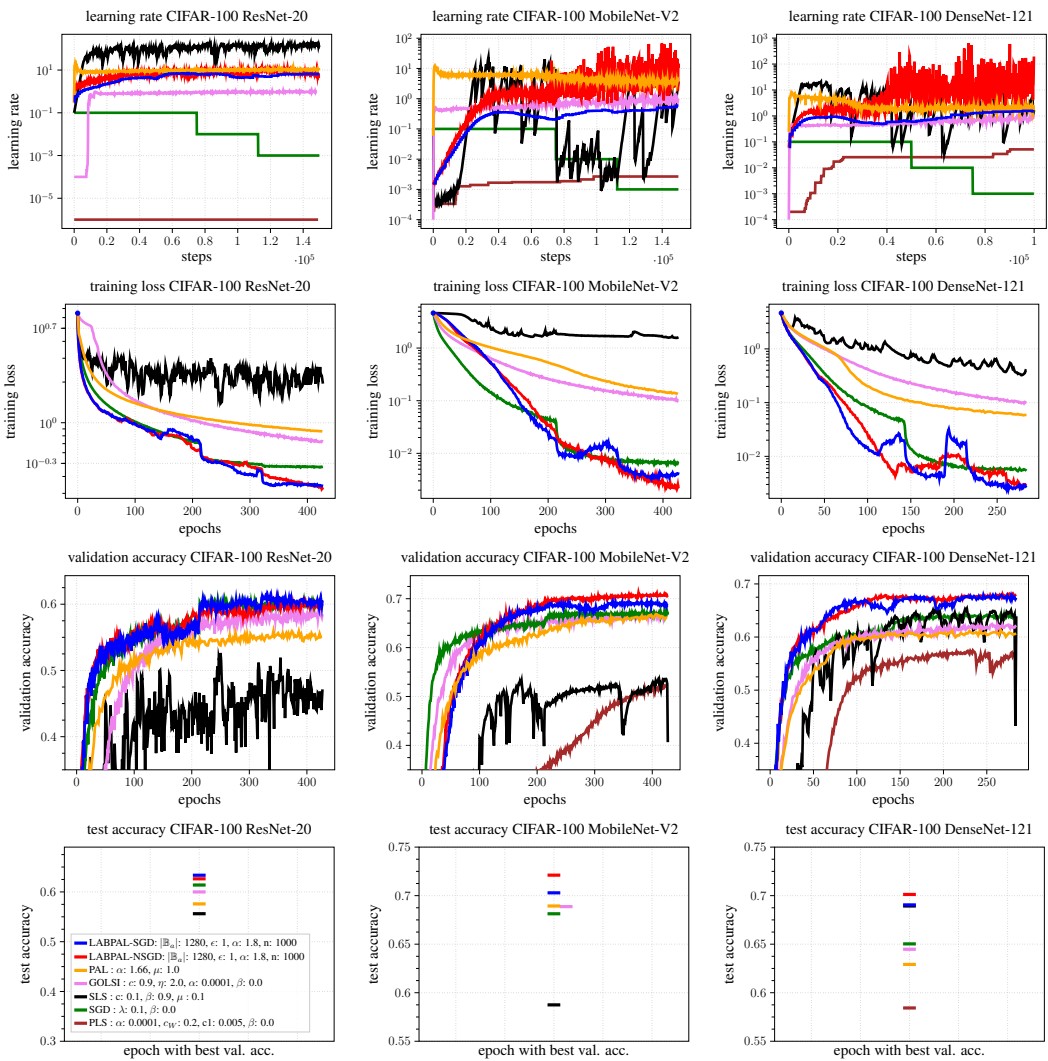

Figure 4: Performance comparison on **CIFAR-100** of our approach LABPAL in the SGD and NSGD variants against several line searches and SGD. Optimal hyperparameters for CIFAR-10 found with a detailed grid search are reused (Appendix G.1). Here, our approaches surpass the other approaches on training loss, validation, and test accuracy. Columns indicate different models. Rows indicate different metrics. Results for CIFAR-10, ImageNet and SVHN are given in appendix Figures 8, 9 and 10. The batch-size used is 128.

Comparison is done across several datasets and models. Specifically, we compare ResNet-20 (He et al., 2016), DenseNet-40 (Huang et al., 2017), and MobileNetV2 (Sandler et al., 2018) trained on CIFAR-10 (Krizhevsky & Hinton, 2009a), CIFAR-100 (Krizhevsky & Hinton, 2009b), and SVHN (Netzer et al., 2011). In addition, we compare MobileNetV3 (Howard et al., 2019) trained on ImageNet (Deng et al., 2009). We concentrate on classification problems since the empirical observations are inferred from a classification task and since those problems are usually considered to benchmark new optimization approaches.

For each optimizer, we perform a comprehensive hyper-parameter grid search over all models trained on CIFAR-10 (see Appendix G.1). The best performing hyper-parameters on the validation set are then reused for all other experiments. The latter is done to check the robustness of the optimizer by handling all other datasets as if they were unknown, as is usually the case in practice. Our aim here is to show that satisfactory results can be achieved on new problems without any fine-tuning needed. Further experimental details are found in Appendix G.

Figure 4 as well as Appendix Figures 8, 10, 9 show that both LABPAL approaches outperform PLS, GOLSI and PAL considering training loss, validation accuracy, and test accuracy. LABPAL&NSGD tends to perform more robust and better than LABPAL&SGD. LABPAL&NSGD outperforms SGD tuned with a piece-wise constant schedule and challenges SLS on validation and test accuracy. On CIFAR-100, our approaches even perform better than all others. The important result is that hyper-parameter tuning for LABPAL is not needed to achieve good results across several models and datasets. However, this also is true for pure SGD, which suggests that the simple rule of performing a step size proportional to $\|\mathbf{g}\|$ is sufficient to implement a well-performing line search. This also strengthens the observation of (Mutschler & Zell, 2021), which states that SGD, with the correct learning rate, is already performing an almost optimal line search.

The derived learning rate schedules of the LABPAL approaches are significantly different from those of the other line search approaches (Figure 4, 8, 10 first row). Interestingly they show a strong *warm up phase* at the beginning of the training followed by a rather *constant phase* which can show minor learning rate changes with an increasing trend. The *warm up* phase is often seen in sophisticated learning rate schedules for SGD; however, usually combined with a *cool down* phase. The latter is not apparent for LABPAL since we increase the batch size. LABPAL&NSGD indirectly uses learning rates of up to $10^7$ but still trains robustly.

A comparison of training speed and memory consumption is given in Appendix D. In short, LABPAL has identical GPU memory consumption as SGD and is on average only $19.6\%$ slower. However, for SGD usually a grid search is needed to find a good $\lambda$, which makes LABPAL considerably cheap.

### 4.3 ADAPTATION TO VARYING GRADIENT NOISE

Recent literature, w.g. (Mutschler & Zell, 2020), (Vaswani et al., 2019), (Kafka & Wilke, 2019) show that line searches work with a relatively large batch size of 128 and a training set size of approximately 40000 on CIFAR-10. *However, a major, yet not deeply considered problem is that line searches operating on the mini-batch loss vary their behavior with another batch- and training set sizes leading to varying gradient noise*. E.g., Figure 5 shows that training with PAL and a batch size of 10 on CIFAR-10 does not work at all. The reason is that the by mini-batches induced gradient noise, and with it the difference between the full-batch loss and the mini-batch loss, increases. However, we can adapt LABPAL to work in these scenarios by holding the *noise scale* it is exposed to approximately constant. As the learning rate is inferred directly, the batch size has to be adapted. Based on the linear approximation of the *noise scale* (see Equation 5), we directly estimate a noise adaptation factor $\epsilon \in \mathbb{R}$ to adapt LABPAL's hyperparameter:

$$\epsilon := \frac{\nu_{new}}{\nu_{ori}} = \frac{|\mathbb{B}_{ori}|}{|\mathbb{B}_{new}|} \frac{|\mathbb{D}_{new}|}{|\mathbb{D}_{ori}|} = \frac{128}{|\mathbb{B}_{new}|} \frac{|\mathbb{D}_{new}|}{40,000} \tag{7}$$

The values the original batch size $|\mathbb{B}_{ori}|$ and the original dataset size $|\mathbb{D}_{ori}|$ originate from our search for best-performing hyperparameters on CIFAR-10 with a training set size of 40,000, a batch size of 128, and 150,000 training steps. We set the number of training steps to $150,000\epsilon$ and multiply the batch sizes in the batch size schedule $k$ by $\epsilon$. This rule makes the approach fully parameter less in practice.

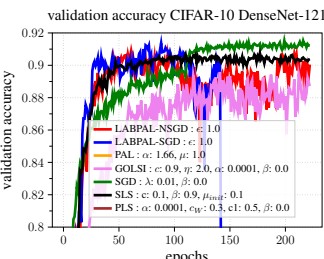 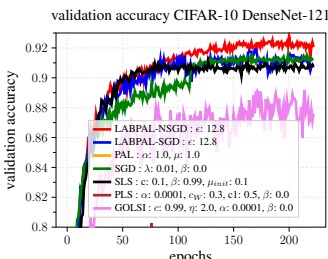

Figure 5: Performance comparison at **batch size 10** on **CIFAR-10**. Left: reusing the same hyper-parameters as for batch size 128. Right: applying the directly estimable noise adaptation factor to the LABPAL approaches and performing a grid search for optimal hyperparameters for the other approaches. On validation and test accuracy the LABPAL approaches outperform SGD, whereas on training loss they compete with it. (For more details see Appendix Figure 11 & 12). PLS curves are incomplete since training failed.

Figure 5 shows that LABPAL approaches fail when the gradient noise factor is not applied, but they improve their performance significantly if it is applied. Now, they even surpass SGD on validation and test accuracy (Appendix Figure 11 & 12). For all other approaches, a comprehensive hyperparameter search is done (see Appendix G.1). Except for SGD, even the best-found hyperparameters could not handle the higher noise.

### 4.4 HYPERPARAMETER SENSITIVITY ANALYSIS

We performed a detailed hyperparameter sensitivity analysis for LABPAL&SGD and LAB-PAL&NSGD. To keep the calculation cost feasible, we investigated the influence of each hyperparameter, keeping all other hyperparameters fixed to the default values (see Algorithm 1). Appendix Figure 6 and 7 show the following characteristics: Estimating new $s_{upd}$ or $\lambda$ with $\mathbb{B}_a$ smaller than 640 decreases the performance since $l_t$ is not fitted well enough (row 1). The performance also decreases if reusing the $\lambda$ (or $s_{upd}$) for more update steps (row 2), and if using a step size adaptation $\alpha$ of less than 1.8 (row 3, except for ResNet). This shows that optimizing for the locally optimal minimum in line direction is not beneficial. From a global perspective, a slight decrease of the loss by performing steps to the other side of the parabola shows more promise. Interestingly, even using $\alpha$ larger than two still leads to good results. (Mutschler & Zell, 2021) showed that the loss valley in line direction becomes wider during training. This might be a reason why these update steps, which should actually increase the loss, work. Using a maximal step size of less than 1.5 (row 7) and increasing the noise adaptation factor $\epsilon$ (row 9) while keeping the batch size constant also decreases the performance. The latter indicates that the inherent noise of SGD is essential for optimization. In addition, we considered a momentum factor and conclude that a value between $0.4$ and $0.6$ increases the performance for both LABPAL approaches (row 5).

## 5 LIMITATIONS

Our approach can only work if the empirically found properties we rely on are apparent or are still a well enough approximation. In Section 4.2 we showed that this is valid for classification tasks. In additional sample experiments, we observed that our approach also works on regression tasks using the square loss. However, it tends to fail if different kinds of losses from significantly different heads of a model are added, as it is often the case for object detection and object segmentation.

A theoretical analysis is lacking since the optimization field still does not know the reason for the local parabolic behavior of $l_t$ is, and consequently, what an appropriate function space to consider for convergence is.

## 6 DISCUSSION & OUTLOOK

This work introduces a robust line search approach for Deep Learning problems based upon empirically found properties of the full-batch loss. Our approach estimates learning rates well across models, datasets, and batch sizes. It mostly surpasses other line search approaches and challenges SGD tuned with a piece-wise constant learning rate schedule. We are the first line search work that analyses and adapts to varying gradient noise. In addition, we show that mini-batch gradient norm information is not necessary for training. In future, we will analyze the causes for the local parabolic behavior of the full-batch loss along lines, to get a better understanding of DNN loss landscapes and especially of why and when specific optimization approaches work.

## REPRODUCIBILITY

Experimental details including all hyperparameters used for the experiments presented in Section 4 are found in App. G.1. The source code to reproduce our experiments including our implementations of SLS, GOLSI, PLS, PAL and LABPAL is provided in the supplementary materials.

## ETHICS STATEMENT

Since we understand our work as basic research, it is extremely error-prone to estimate its *specific* ethical aspects and future positive or negative social consequences. As optimization research influences the whole field of deep learning, we refer to the following works, which discuss the ethical aspects and social consequences of AI and Deep Learning in a comprehensive and general way: Yudkowsky et al. (2008); Muehlhauser & Helm (2012); Bostrom & Yudkowsky (2014).

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

# A HYPERPARAMTER SENSITIVITY ANALYSIS

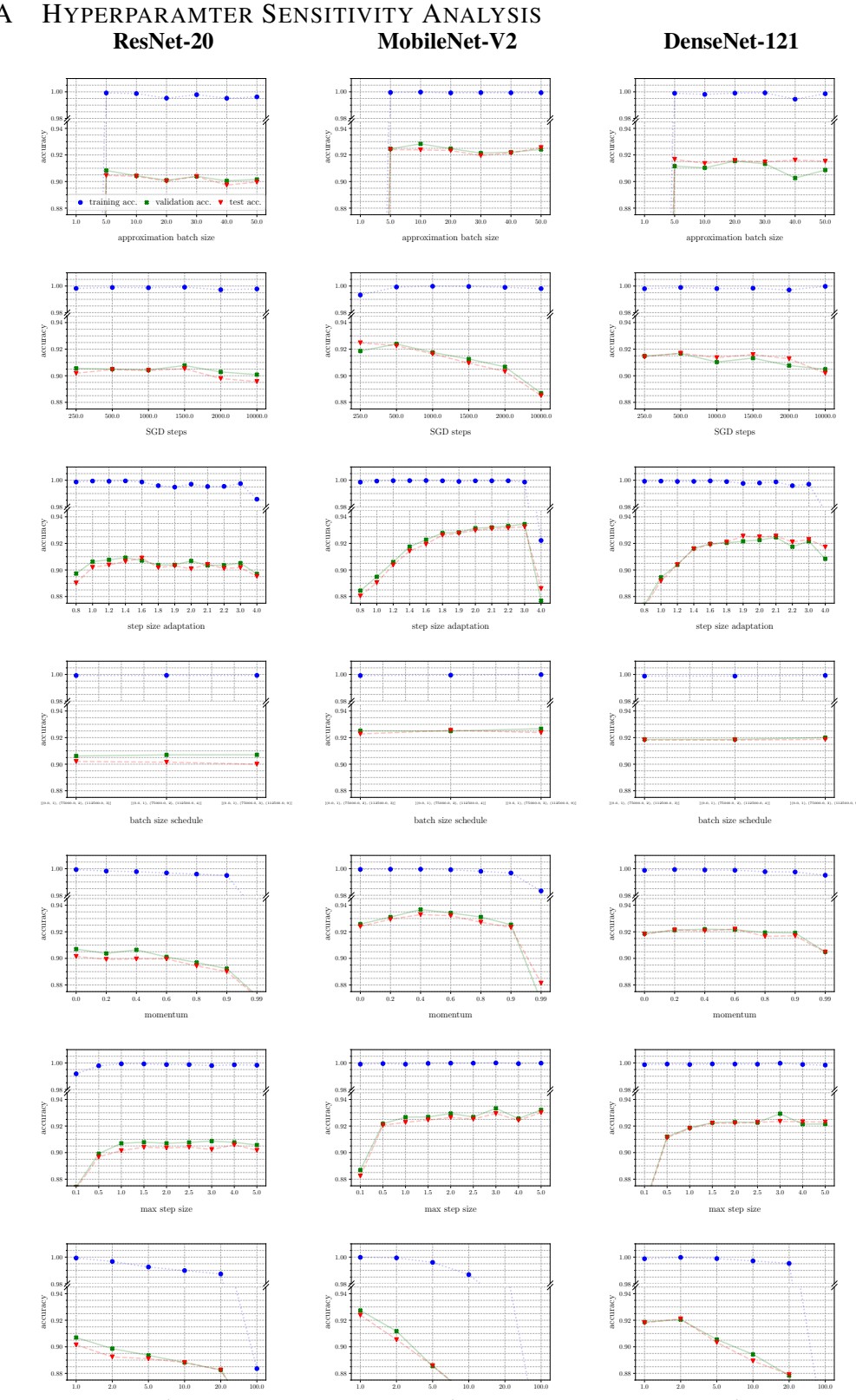

Figure 6: Sensitivity analysis of parameters of **LABPAL&SGD**. The observations of LAB-PAL&NSGD described in Figure 7 are also valid for LABPAL&NSGD.

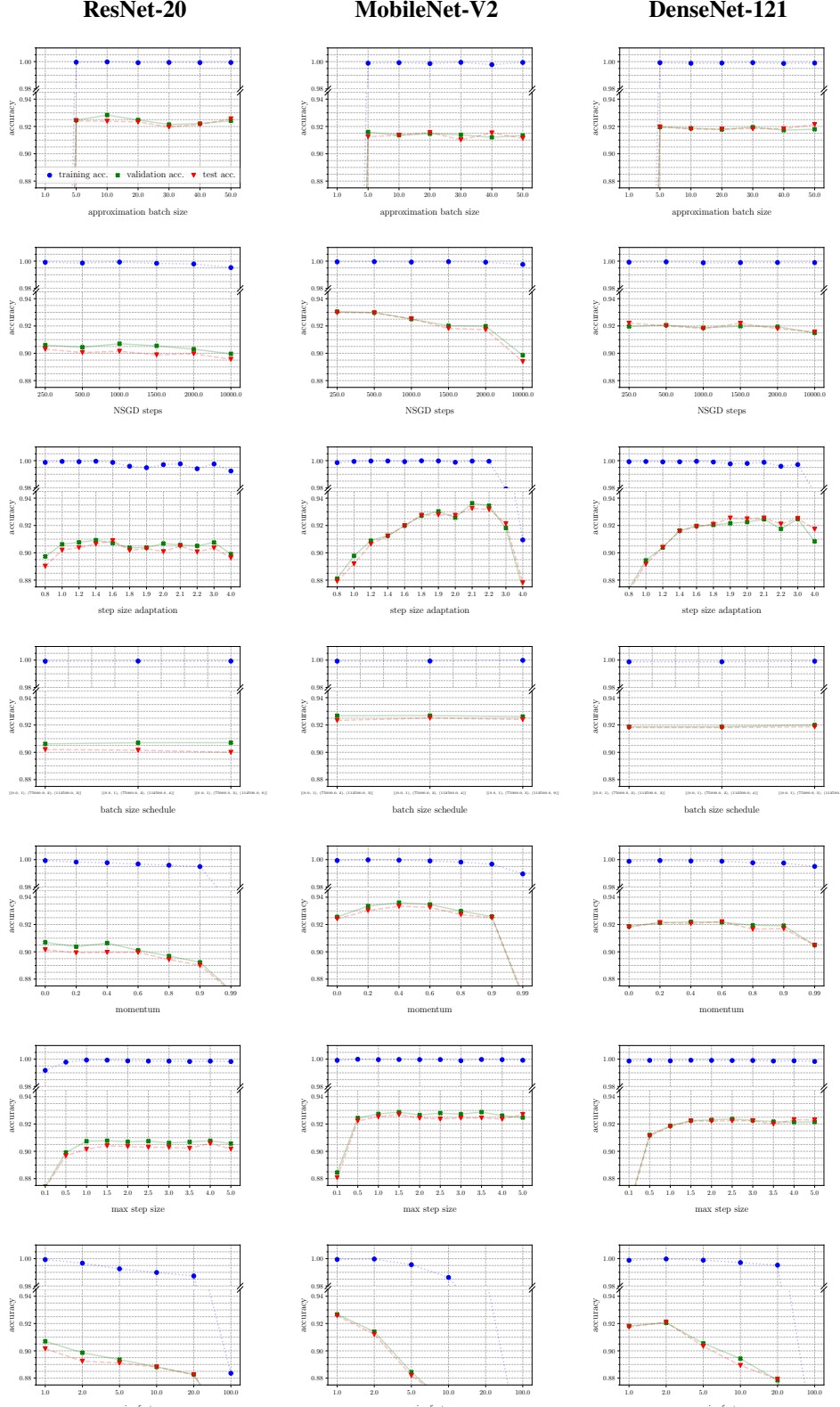

Figure 7: Sensitivity analysis of parameters of **LABPAL&NSGD**. The default parameters are: approximation batch size $\mathbb{B}_a = 1280$, SGD steps $s = 1000$, step size adaptation $\alpha = 1.8$, batch size schedule $k = (0{:}1, 75000{:}2, 112500{:}4)$, momentum $\beta = 0$, maximal step size $= 1.0$, noise-factor $\epsilon = 1$. For $\mathbb{B}_a$ the factor 128 is multiplied with is given on the x axis.

# B FURTHER PERFORMANCE COMPARISONS

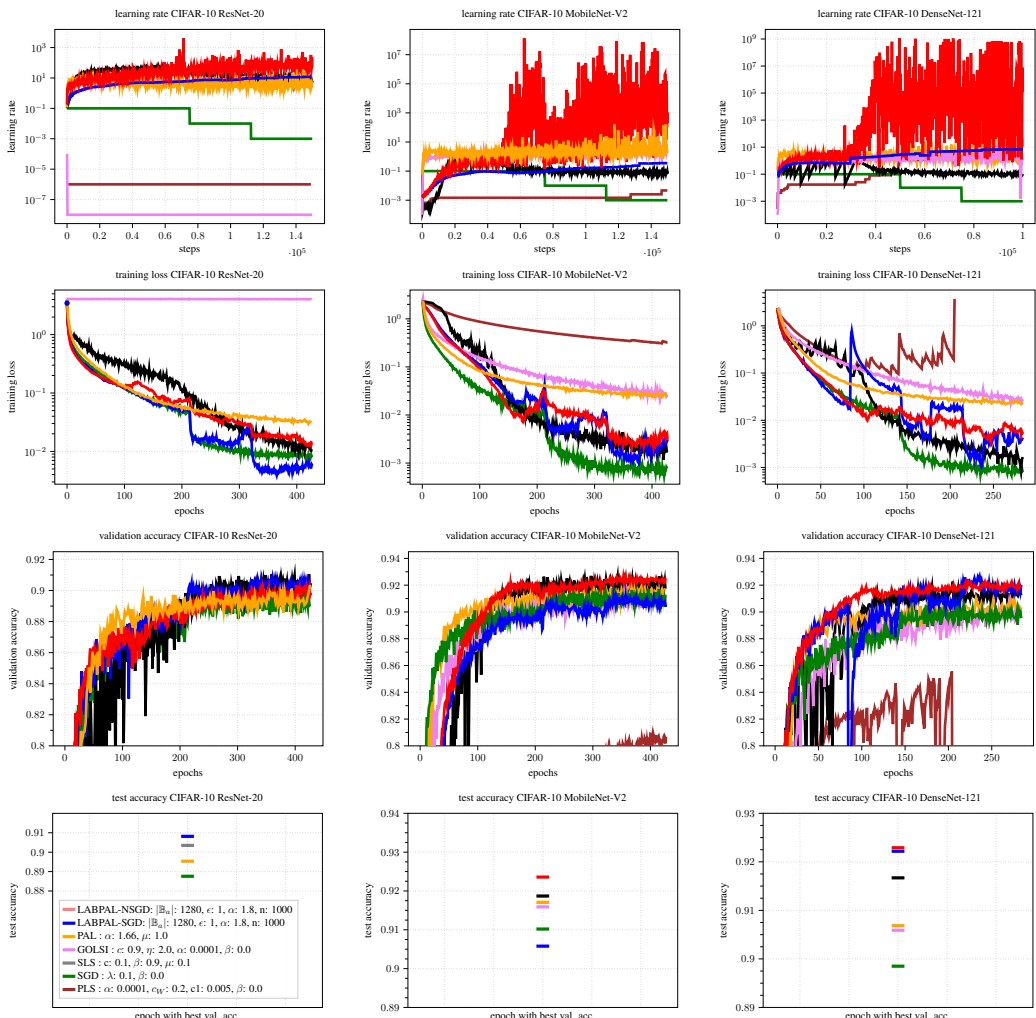

Figure 8: Performance comparison on **CIFAR-10** of our approach LABPAL in the SGD and NSGD variants against several line searches and SGD. Optimal hyperparameters are found with an elaborate grid search. Our approaches challenge and often outperform the other approaches on training loss, validation, and test accuracy. Columns indicate different models. Rows indicate different metrics.

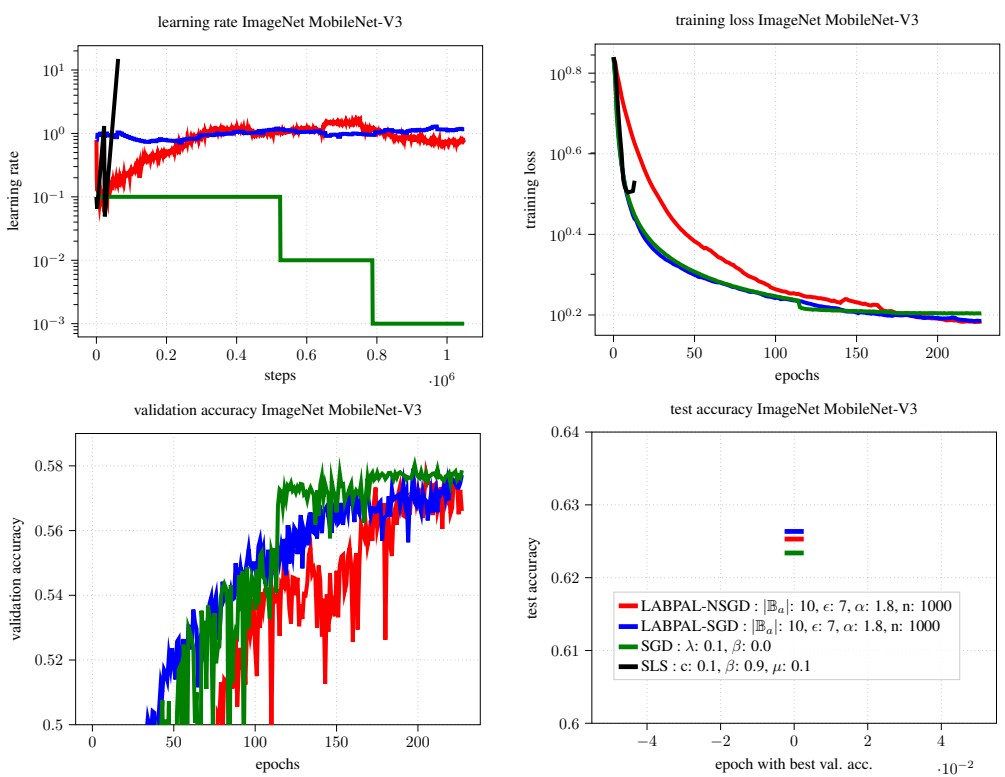

Figure 9: Performance comparison of top 1 error on **IMAGENET** of our approach LABPAL in the SGD and NSGD variants against SLS and SGD. Optimal hyperparameters are found with an elaborate grid search. Optimal hyperparameters found with a detailed grid search for CIFAR-10 are reused. Our approaches challenge the other approaches on training loss and test accuracy. SLS fails shortly after the beginning of the training due to too high estimated learning rates. For LABPAL the adaptation factor $\epsilon$ introduced in Section 4.3 is applied.

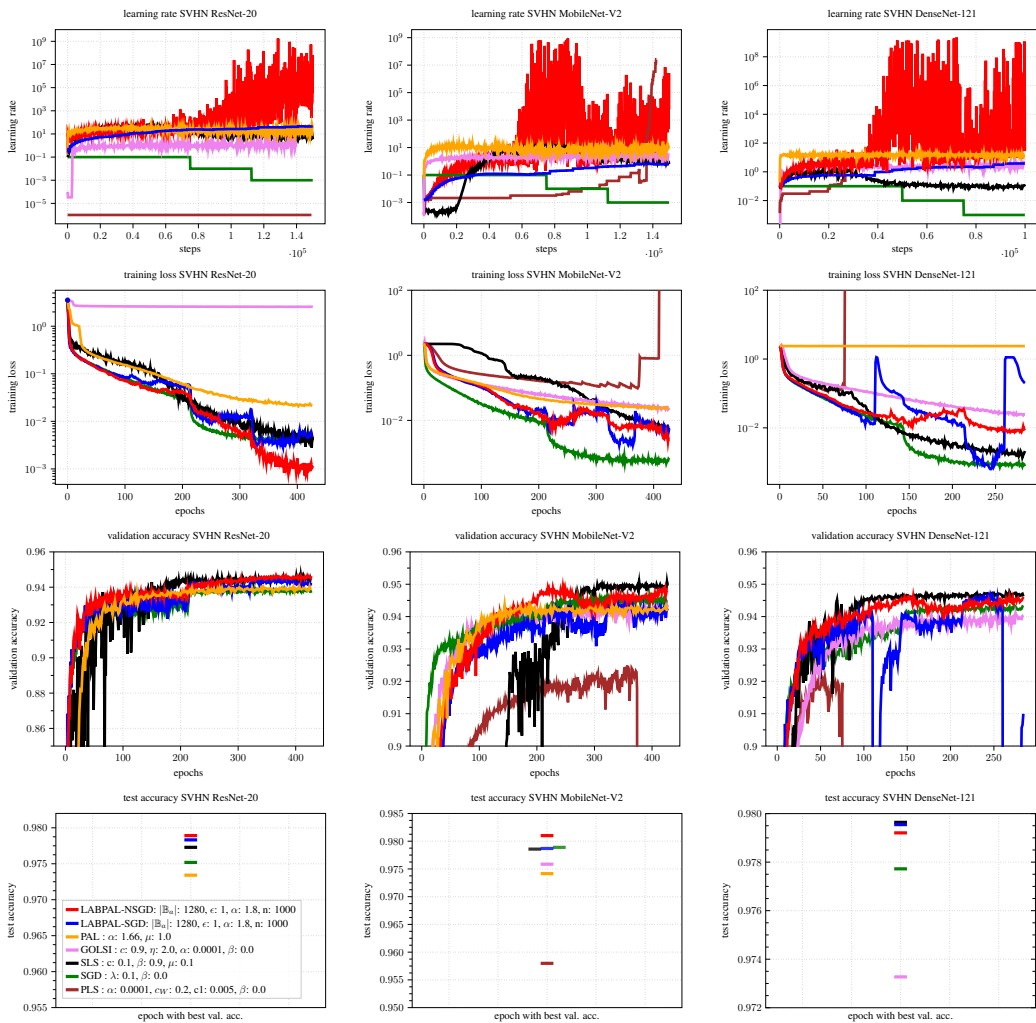

Figure 10: Performance comparison on **SVHN** of our approach LABPAL in the SGD and NSGD variants against several line search and SGD. Optimal hyperparameters found with a detailed grid search for CIFAR-10 are reused. Our approaches challenge and often surpass the other approaches on training loss, validation, and test accuracy. Columns indicate different models. Rows indicate different metrics.

## C FURTHER RESULTS FOR BATCH SIZE 10

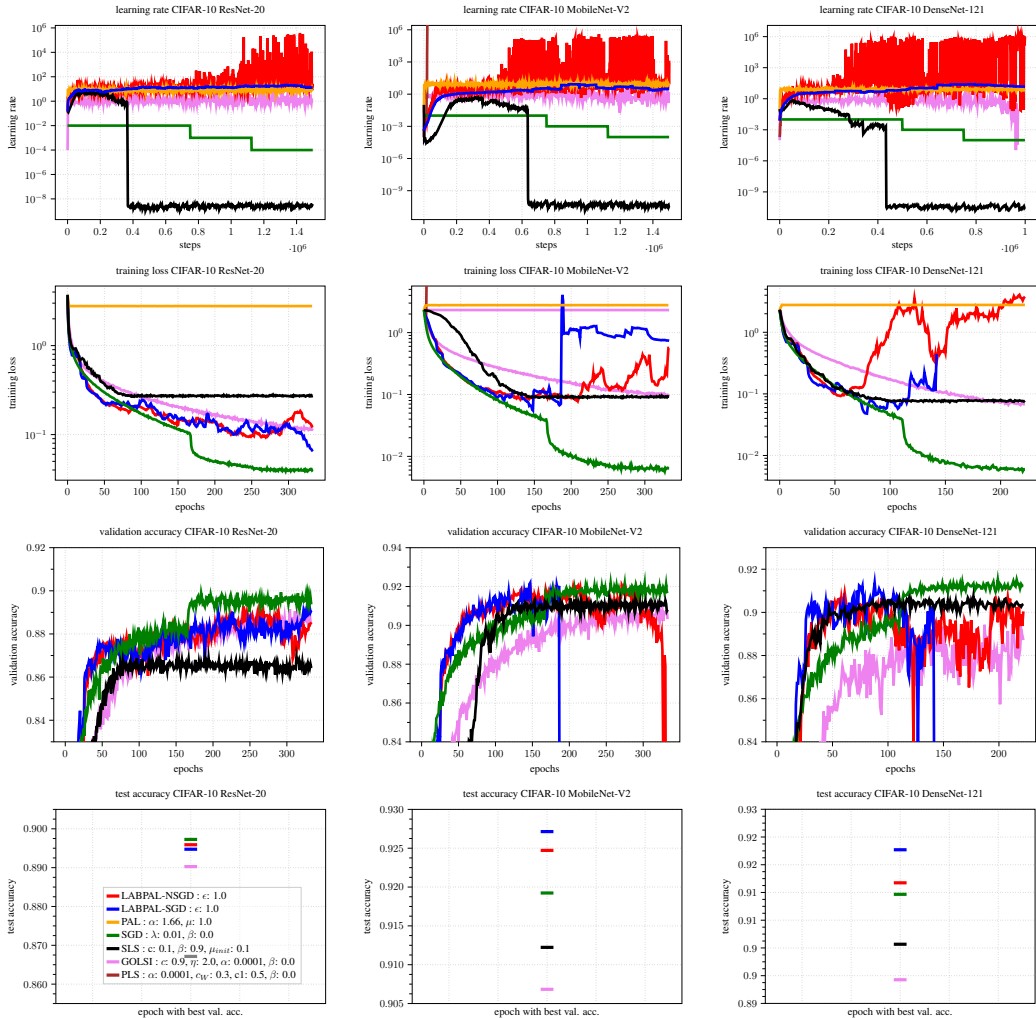

Figure 11: Performance comparison of several models on CIFAR-10 with **batch size 10**. **The same hyperparameters are used as for batch size 128** (see Figure 8). PAL and PLS fail in this scenario. The LABPAL approaches work well in the beginning but fail to estimate well learning rates later; this is solved in Figure 12. Interestingly, they still achieve competitive validation and test accuracies.

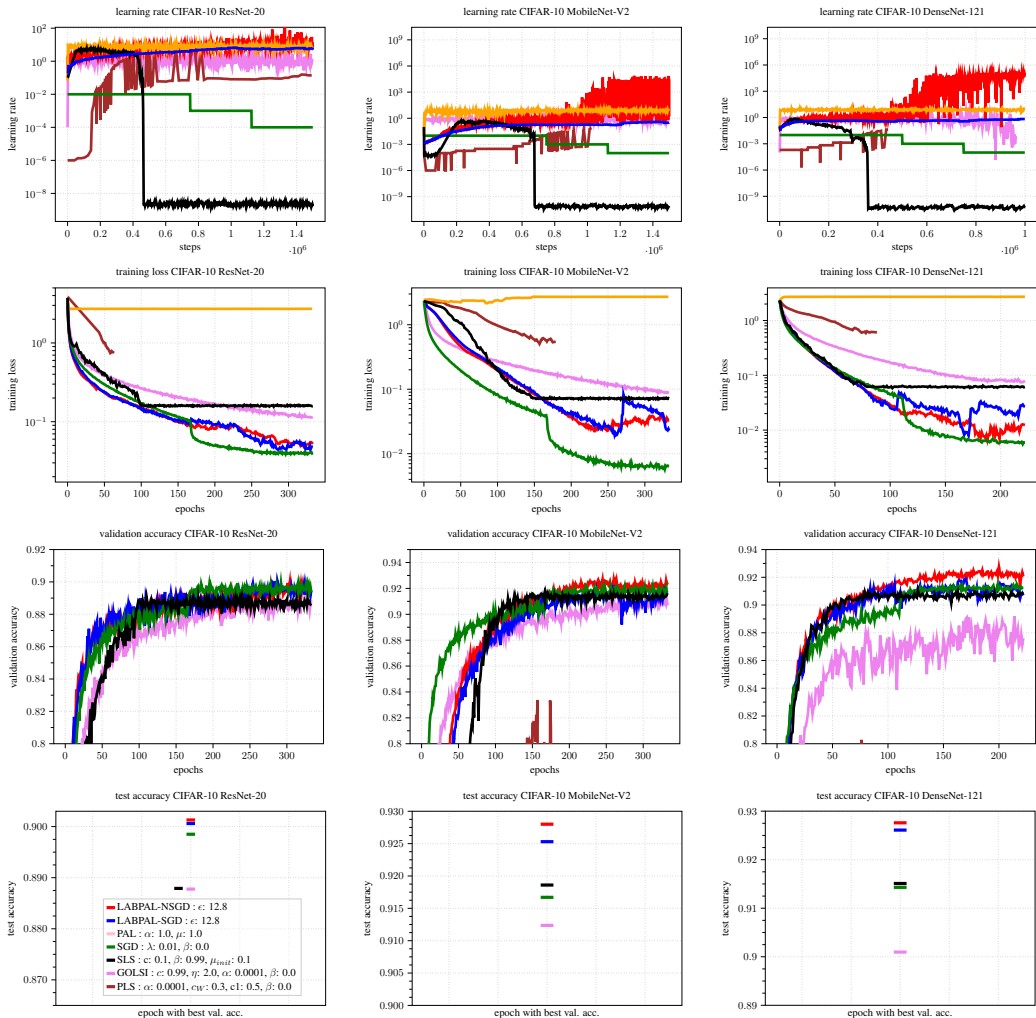

Figure 12: Performance comparison of several models on CIFAR-10 with **batch size 10**. For the LABPAL approaches only the noise factor is adapted according to equation 7. For all other approaches, a grid search is performed to find the best hyperparameters for this scenario. (see Appendix G.1). In comparison to Figure 11, now, the LABPAL approaches perform competitive on the training loss and surpass the other approaches on validation and test accuracy. PLS plots are incomplete since the training failed after some steps.

# D WALL CLOCK TIME AND GPU MEMORY COMPARISON

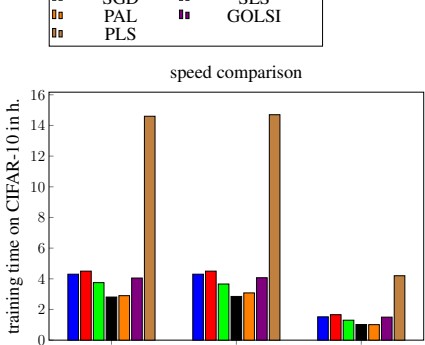
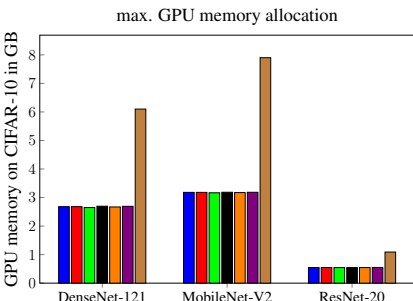

Figure 13: **Left:** Training time comparison on CIFAR-10. SGD, SLS, and PAL show similar training times. GOLSI, and both variants of LABPAL are slightly slower (up to 19.6%). However, a slightly longer training time is acceptable if less time has to be spent in hyper-parameter tuning. PLS is significantly slower. Note that in comparison to SGD, the implementations of the other optimizers are not optimized on CUDA level. **Right:** Maximum allocated memory comparison on CIFAR-10. Except for PLS all approaches need approximately the same amount of memory.

# E THEORETICAL CONSIDERATIONS

As the field does not know what the reason for the local parabolic behavior of the full-batch loss is and, thus, what an appropriate function space to consider for convergence is, we refer to the theoretical analysis of (Mutschler & Zell, 2020). They show convergence on a quadratic loss. This is also valid for LABPAL, with the addition that each mini batch-loss can be of any form as long as the mean over these losses is a quadratic function.

# F RELATION OF UPDATE STEP ADAPTATION $\alpha$ AND THE FIRST WOLFE CONSTANT $w_1$.

Let $f : \mathbb{R} \to \mathbb{R}$ be of form $x \mapsto ay^2 + by + c$. We start with the first Wolfe condition (a.k.a. Armijo condition, sufficient decrease condition):

$$f(x_0 + y) \leq f(x_0) - y\nabla f(x_0)w_1 \qquad \text{in our case } x_0 = 0, w_1 \text{ wolfe constant} \tag{8}$$
$$f(y) \leq f(0) + ybw_1 \tag{9}$$
$$ay^2 + by + c \leq c + ybw_1 \qquad \text{use quadratic shape, } \nabla f(x_0) = b \tag{10}$$
$$ay^2 + by - ybw_1 \overset{!}{=} 0 \tag{11}$$
$$\frac{ay^2 + by}{by} = \frac{ay}{b} + 1 = w_1 \tag{12}$$
$$-\frac{\alpha}{2} + 1 = w_1 \qquad \text{set } y = \alpha\frac{-b}{2a}, \alpha \in [1, 2) \tag{13}$$
$$-2w_1 + 2 = \alpha \tag{14}$$

## G  FURTHER EXPERIMENTAL DETAILS

Further experimental details for the optimizer comparison in Figure 8,4,10,5,11,12 of Sections 4.2 & 4.3.

**PLS:** We adapted the only available and empirically improved TensorFlow (Abadi et al., 2015) implementation of PLS (Balles, 2017), which was transferred to PyTorch (Paszke et al., 2019) by (Vaswani et al., 2019), to run on several state-of-the-art models and datasets.

The training steps for the experiments in section Section 4 were 100,000 for DenseNet and 150,000 steps for MobileNetv2 and ResNet-20. Note that we define one training step as processing one input batch to keep line search approaches comparable.

The batch size was 128 for all experiments. The validation/train set splits were: 5,000/45,000 for CIFAR-10 and CIFAR-100 20,000/45,000 for SVHN.

All images were normalized with a mean and standard deviation determined over the dataset. We used random horizontal flips and random cropping of size 32. The padding of the random crop was 8 for CIFAR-100 and 4 for SVHN and CIFAR-10.

All trainings were performed on Nvidia Geforce 1080-TI GPUs.

Results were averaged over three runs initialized with three different seeds for each experiment.

For implementation details, refer to the source code provided in the supplementary materials .

### G.1  HYPERPARAMETER GRID SEARCH ON CIFAR-10

For our evaluation, we used all combinations out of the following hyperparameters.

*SGD*:

| hyperparameter | symbol | values |
|---|---|---|
| learning rate | $\lambda$ | $\{0.1, 0.01, 0.001, 0.0001\}$ |
| momentum | $\alpha$ | $\{0.0\}$ |

*PAL*:

| hyperparameter | symbol | values |
|---|---|---|
| measuring step size | $\mu$ | $\{0.1, 1\}$ |
| direction adaptation factor | $\beta$ | $\{0.0\}$ |
| update step adaptation | $\alpha$ | $\{1, 1.66\}$ |
| maximum step size | $s_{max}$ | $\{3.16 \ (\approx 10^{0.5})\}$ |

*LABPAL (SGD and NSGD)*:

| hyperparameter | symbol | values |
|---|---|---|
| step size adaptation | $\alpha$ | $\{1.0, 1.25, 1.5, 1.8, 1.9\}$ |
| SGD steps | $\beta$ | $\{1000\}$ to keep speed comparable |
| initial measuring step size | $\gamma$ | $\{0.01, 0.1\}$ |
| parabolic approximation sample step size | | $\{0.1, 0.01\}$ |
| approximation batch size | $b$ | $\{5, 10, 20\}$ |
| batch size schedule | $bs$ | $\{\{0 : 1, 75000 : 2, 112500 : 3\}, \{0 : 1, 75000 : 2, 112500 : 4\}\}$ |

*GOLSI*:

| hyperparameter | symbol | values |
|---|---|---|
| initial step size | $\mu$ | $\{0.1, 1\}$ |
| momentum | $\beta$ | $\{0.0\}$ |
| step size scaling parameter | $\eta$ | $\{0.2, 2.0\}$ |
| modified wolfe condition parameter | $c2$ | $\{0.9, 0.99\}$ |

*PLS*:

| hyperparameter | symbol | values |
|---|---|---|
| first wolfe condition parameter | $c_1$ | $\{0.005, 0.05, 0.5\}$ |
| acceptance threshold for the wolfe probability | $cW$ | $\{0.2, 0.3, 0.4\}$ |
| initial step size | $\alpha_0$ | $\{10^{-4}\}$ |

*SLS*:

| hyperparameter | symbol | values |
|---|---|---|
| initial step size | $\mu$ | $\{0.1, 1\}$ |
| step size decay | $\beta$ | $\{0.9, 0.99\}$ |
| step size reset | $\gamma$ | $\{2.0\}$ |
| Armijo constant | $c$ | $\{0.1, 0.01\}$ |
| maximum step size | $\mu_{max}$ | $\{10.0\}$ |

