# OpenReview forum: "Using a one dimensional parabolic model of the full-batch loss to estimate learning rates during training"
_ICLR.cc/2022/Conference — ICLR 2022 Submitted_

### Official Review · Reviewer_mG5J · 2021-11-02

**Correctness:** 3
**Technical Novelty And Significance:** 2
**Empirical Novelty And Significance:** 2
**Recommendation:** 3
**Confidence:** 4

**Main Review:**

In this paper, the authors propose an algorithm for the automatic selection of the step size for SGD and apply this algorithm for DNN training. They suggest that full batch loss along the negative direction of the normalized batch gradient could be localy approximated with a parabola with high precision. So optimal step size could be also approximated as distance to minimum of that parabola. The authors notice that optimal step size changes slowly during SGD optimization process therefore one can reuse optimal step size for number of consequtive SGD updates and by doing that reduce algorithm complexity and made it applicable.

Weak points:

-W1. The overall novelty of paper is limited because propesed method LABPAL is straightforward modification of previously published PAL algorithm (https://arxiv.org/pdf/1903.11991.pdf). In LABPAL full batch loss is approximated via multiple  batches compared to single batch approximation in PAL. The difference in a way to perform parabolic approximation seems like a minor modification. Also, the experimental background for applicability of parabolic approximation (i.e. Observations 1, 2, 3, 4, 5, 6) is basically stated in the same way in https://arxiv.org/pdf/2103.17132.pdf as well as illustrations to these observations (figures 1, 2 matches figures 2, 6 from the latter paper). Other technical details like adaptation to varying gradient noise and relation with first Wolfe constant have marginal influence on the main point of the paper.
-W2. The Observation 2 seems not well reasoned since the conclusion is made for only one choice of dataset and model and in other settings this observation may fail.

Strong points:

-S1. The experiments show that the proposed method outperforms other baseline methods on various datasets and that LABPAL produce stable results for a wide range of hyperparameters.


**Summary Of The Paper:**

In this paper, the authors propose an algorithm for the automatic selection of the step size for SGD and apply this algorithm for DNN training. They suggest that full batch loss along the negative direction of the normalized batch gradient could be localy approximated with a parabola with high precision. So optimal step size could be also approximated as distance to minimum of that parabola. The authors notice that optimal step size changes slowly during SGD optimization process therefore one can reuse optimal step size for number of consequtive SGD updates and by doing that reduce algorithm complexity and made it applicable.

**Summary Of The Review:**

It seems that most part of the paper (including figures and statements) is taken from one of two previously published papers and hence the novelty contained in the paper itself is very limited.  This inclines me towards rejection of the paper in its current form.

---

### Official Review · Reviewer_kiNL · 2021-11-03

**Correctness:** 2
**Technical Novelty And Significance:** 2
**Empirical Novelty And Significance:** 2
**Recommendation:** 3
**Confidence:** 4

**Main Review:**

Strengths:
To the best of my knowledge, the exact idea proposed in this paper is novel.

Weaknesses:
- The algorithm is built on empirical observations of a specific image classification workload. The experiments only include image classification workloads, so whether this algorithm generalizes to other tasks (e.g. NLP tasks) is an open question.
- Many of the design decisions made for the algorithm seem arbitrary, and there are many additional hyperparameters that seem important to tune. The method 1) approximates the full-batch loss using many mini-batch evaluations, 2) reuses the same step size for some consecutive steps, 3) increases the batch size with a piecewise constant schedule as training progress, and 4)uses a slightly larger step size value than the actual optimum given by the parabola by multiplying by a factor between 1 and 2. There are no ablation studies to understand which of these choices are important. Furthermore, they introduce many additional hyperparameters. In fact, looking at Section G.1 in the Appendix, it seems like LABPAL has the most number of hyperparameters.
- There is no comparison to SGD with Momentum, which performs the best in the workloads considered in the experiments. Given that the method has so many additional hyperparameters, and the comparison with the best-performing optimizer is lacking, the practicality of the proposed method is questionable.
- Section 4.3 (adaptation to varying gradient noise) doesn’t make much sense to me. It seems like all that is happening is the batch size is being increased. So then the comparison with other methods doesn’t seem fair, because the batch size is no longer the same.
- Convergence analysis is missing.

Comments and questions:
- The distinction between step size and learning rate is unclear. What exactly is the difference? Could you make things clearer in the paper? For example, it seems like in Equation 6, learning rate = \lambda = alpha * s_{min,t}/||g_t||, but in the algorithm box, it seems like learning rate = learning rate * alpha.
- In Section G.1, there are some parameters under LABPAL that I’m not sure were mentioned in the main paper: initial measuring step size, parabolic approximation sample step size, and approximation batch size.

----
Update:
I have read the reviews from the other reviewers and the author responses. I agree with the authors that the paper's contributions are real and valid. However, as other reviewers also noted, I don't think the contributions are enough for acceptance, therefore, I am keeping my score. I would be excited to see a future version of the paper that has a wider range of experiments (maybe showing that the properties also hold for other deep learning tasks).

**Summary Of The Paper:**

This paper uses empirical observations of Mutschler & Zell (2021) to come up with a line search algorithm called LABPAL, which is essentially fitting a 1-D parabola to the (approximate) full-batch loss in the direction of the mini-batch gradient, and using the minima of the parabola to obtain the step size. The authors include various tricks to make this approach work in practice. The show empirically on various image classification workloads that LABPAL is mostly better or on-par with SGD (with piecewise constant learning rate schedule) and other recent line-search methods.


**Summary Of The Review:**

The proposed method doesn’t seem practically useful. It introduces many additional hyperparameters, and it is unclear whether it performs better than SGD with Momentum, and in settings where it is not confirmed whether the empirical observations that justify the method hold. Therefore, I recommend rejection.

---

### Official Review · Reviewer_rHNN · 2021-11-04

**Correctness:** 2
**Technical Novelty And Significance:** 2
**Empirical Novelty And Significance:** 2
**Recommendation:** 3
**Confidence:** 2

**Main Review:**

Adaptive learning rate methods for SGD are important and improvements over previous line search methods have the potential for significant impact.  However, the evaluation of different methods in the paper are either lacking or significantly suboptimal.

For the former, evaluated methods should be more extensive; SGD with a constant learning rate or piece-wise constant learning rate are not widely used (outside of ResNet architectures for the latter [1]).  At the very least, SGD with a decaying learning rate should be compared to.  Furthermore, the evaluation of other methods does not seem correct.  For instance, SLS uniformly performs terribly for CIFAR-100 ResNet and DenseNet, in stark contrast to the results in the original SLS NeurIPS 2019 paper.  Can the authors please comment on such a large discrepancy?  With regards to how the hyperparmaters for various methods are tuned, the grid search looks coarse and the search space very suboptimal, which is supported by ResNet-20 only achieving ~60% accuracy (for SGD) on CIFAR-100.  This is further supported by PLS only beginning with an initial step size of 10^{-4}; ResNet architectures for CIFAR-100 have displayed excellent performance using PLS with an initial learning rate of 0.1, which is then divided by 10 half-way through training and further divided by 10 three-fourths through training [1].  At the very least, the initial learning rate for this optimizer should always be the optimal learning rate for SGD output during hyperparameter optimization.

Other comments:
-"Gradient-only line search (GOLSI)... its performance is rather weak."<- Please include citation for this claim

-"This was done for 10,000 consecutive SGD update steps of a ResNet18’s training process on a subset of CIFAR-10. Representative plots of their 10,000 measured full-batch losses along lines are presented in Figure 1. Relevant insights and found properties of these works will be introduced and exploited to derive our algorithm in Section 3." <- The problem with basing all presented work on this single study is that different learning schedules for SGD (and
variants) differ significantly for different datasets and problem domains.  See:
Schmidt, Robin M., Frank Schneider, and Philipp
Hennig. "Descending through a crowded valley-benchmarking deep
learning optimizers." International Conference on Machine
Learning. PMLR, 2021.

-"Besides decreasing the learning rate, increasing the batch size remains an important choice to tackle gradient noise" <- This is a point of contention in the literature; several studies report that increasing the batch size degrades performance [2,3].

-The observation/derivation section in the paper is very difficult to read.

-To clarify, the authors shrink the batch size from 128 to 10 in Figure 5, as a surrogate for increasing gradient noise.  Is that correct?  It would be interesting to see intermediate batch sizes from 128 decreasing to 10 to actually see how noise floor increases w.r.t. mini-batch size (a batch size of 10 is unreasonably small).


References:
[1] He, Kaiming, et al. "Identity mappings in deep residual networks."
European conference on computer vision. Springer, Cham, 2016.
[2] Keskar, Nitish Shirish, et al. "On large-batch training for deep learning: Generalization gap and sharp minima." arXiv preprint arXiv:1609.04836 (2016).
[3] Goodfellow, Ian, Yoshua Bengio, and Aaron Courville. Deep learning. MIT press, 2016.


**Summary Of The Paper:**

The authors propose a parabolic line search for SGD to automatically adjust the learning rate during training.  The authors emphasize that their method is not motivated by theory, but on a previous empirical evaluation of ResNet-20 on CIFAR-10.  Their method is evaluated against several other recently proposed line search strategies for SGD (along with SGD using constant and piecewise learning rates) on CIFAR-100 for Resnet-20, MobileNet-V2, and DenseNet-121 (all using a mini-batch of 128).  The authors further show (in their method) how to overcome increasing noise levels in the gradient estimation of SGD by considering a mini-batch size of 10.

**Summary Of The Review:**

Interesting take on a previously explored strategy for adaptable learning rates in SGD.  However, the results and (lack of) comparison to previous related methods does not look correct.

---

### Official Review · Reviewer_w2rC · 2021-11-04

**Correctness:** 4
**Technical Novelty And Significance:** 2
**Empirical Novelty And Significance:** 2
**Recommendation:** 5
**Confidence:** 4

**Main Review:**

The main drawback of this work is that LABPAL is compared to SGD and not to SGD with momentum, even though the fundamental idea behind them is very similar. I think what could make this work strong is showing a competitive result by comparing Momentum SGD with and without LABPAL step sizes. Without it the practical value of the proposed idea is quite limited even if it may outperform other line search methods or SGD tuned optimally.

Another concern I have is in the limited technical novelty and its implication when taking into account other scenarios where the assumptions/observations do not hold or generalize. LABPAL is essentially the backtracking line search with estimating full loss with multiple large mini-batches, and yet its reliance on empirical observations only pertaining to a certain task (image classification) make the scope of this method being applied to quite limited, not to mention that the arbitrarily chosen hyperparameters required to make LABPAL work may not generalize.

The Derivation step 4 seems really like a heuristics that may not hold depending on some data distributions/characteristics.

It's not obvious that why the step sizes have to increase after some period of time.

**Summary Of The Paper:**

This work proposes a line search method LABPAL for first order methods to train neural networks with SGD. This work is motivated to address the issues in previous works that while the classical line search methods assume the exact descent direction, which is expensive to calculate for large data, the inexact methods based on mini-batches of examples are not robust. The idea is to leverage the two empirical observations on the loss landscape that the full-batch loss along lines in the SGD update direction is shaped parabolically and the optimal step size changes rather slowly. The authors provide extensive experimental results using various neural network models for image classification to compare LABPAL performed with either SGD or normalized SGD to other line search methods and show that LABPAL can perform better than SGD tuned optimally.

**Summary Of The Review:**

While it is interesting to see a new line search method that is competitive to existing line search methods, the comparison doesn't seem entirely fair given that the proposed method shares essentially the same idea used in momentum. Maybe authors could give an estimate as to the required amount of total computations and compare it to that of momentum SGD.

---

### Decision · Program_Chairs · 2022-01-20

**Decision:**

Reject

**Comment:**

This work proposes a method for automatic adaptation of the learning rate via a estimating quadratic approximation of the full batch during training. The method motivated by two observed properties of the loss landscape, first the full batch loss along the gradient direction is well approximated by a quadratic polynomial, and second the optimal full batch step size does not change quickly during training. Two primary criticisms raised by reviewers is the weak experimental evidence provided to validate the method and similarities with other approaches for adapting the learning rate. Ultimately reviewers remain unconvinced by the rebuttal and maintained their scores. The AC further stresses the difficulty of properly (and fairly) comparing optimization methods in deep learning. As is consistently shown in the literature, optimizer performance is typically dominated by hyperparameter tuning, this is particularly problematic when submissions tune their own baselines as authors naturally are incentivized to tweak their own methods until the method looks favorable relative to others. Comparing directly against prior published results tuned by other researchers would help alleviate reviewer concerns regarding hyperparameter tuning.